# Quantitative Measurements of Vessel Density and Blood Flow Areas Primary Angle Closure Diseases: A Study of Optical Coherence Tomography Angiography

**DOI:** 10.3390/jcm11144040

**Published:** 2022-07-13

**Authors:** Bingying Lin, Chengguo Zuo, Xinbo Gao, Danping Huang, Mingkai Lin

**Affiliations:** State Key Laboratory of Ophthalmology, Zhongshan Ophthalmic Center, Sun Yat-sen University, Guangzhou 510060, China; linby5@mail2.sysu.edu.cn (B.L.); zuochengguo@mail.sysu.edu.cn (C.Z.); gaoxb@mail.sysu.edu.cn (X.G.)

**Keywords:** PACD, PACG, OCTA, vessel densities, blood flow area

## Abstract

(1) Purpose: To measure the change in vessel density (VD) and the flow area (FA) on the retina of eyes with primary angle-closure diseases (PACD), including primary angle-closure suspect (PACS), primary angle-closure (PAC), acute primary angle-closure (APAC) and primary angle-closure glaucoma (PACG). (2) Methods: Patients with PACD were prospectively enrolled in this study. All participants underwent thorough ophthalmic examinations. The mean defect (MD), retinal nerve fiber layer (RNFL), ganglion cell complex (GCC) thickness, VD measurement, and blood FA were measured. (3) Results: A total of 147 eyes from 121 subjects were included in this study. The VD of the nerve head layer was significantly lower in PACG and APAC (all *p* < 0.001). APAC and PACG had lower FA of all layers, except for the choroid layer (*p* < 0.05). The macular VD of the whole image and blood FA in the superficial layer was significantly lower in PACG (all *p* < 0.001). The MD, RNFL, and GCC thickness demonstrated a strong correlation with whole image VD in the superficial layer (*p* < 0.001), while the inside disc VD did not show a significant correlation with MD, RNFL, and GCC thickness (*p* > 0.05). (4) Conclusions: There was a significant decrease in the VD and FA on the optic disc as well as the VD and FA of the superficial layer in the macular area in APAC and PACG. The changes in VD and FA are correlated with the severity of the glaucomatous structural damage and functional impairment.

## 1. Introduction

Glaucoma is one of the leading causes of irreversible blindness worldwide. PACG is a major form of glaucoma, especially among Asian populations [1]. According to a survey conducted in 2014, the number of people with PACG worldwide is predicted to reach approximately 23.4 million in 2020 and 32.0 million in 2040, 76.7% of whom will be Asian [2]. The pathogenesis of PACG is yet unclear. An important theory of glaucomatous optic neuropathy (GON) is “the vascular theory” [3]. The vascular theory suggests that glaucoma can cause a decrease in the blood flow to the optic nerve head (ONH) [4]. As proposed by the International Society of Geographical and Epidemiological Ophthalmology (ISGEO), PACG belongs to primary angle-closure disease (PACD), which also includes primary angle-closure suspect (PACS) and primary angle-closure (PAC) [5].

Most previous studies have demonstrated a reduced ONH vessel density in patients with POAG [6,7,8,9,10,11]. However, only a few studies have investigated vascular factors in PACG [12,13,14]. To our knowledge, there are few relevant studies concerning the vascular pathogenesis of PACD [15]. Whether insufficient vascular supply also participates in the pathogenesis of PACD needs to be further explored.

Optical coherence tomography angiography (OCTA) is a newly developed technology that can provide highly detailed and layer-specific views of retinal and disc microcirculation in vivo that clearly demonstrate the vascular bed distribution [9,16]. With OCTA, it is easy to obtain important information about the perfusion of the retina and disc.

The purpose of the current study was to investigate the vessel density (VD) and blood flow area (FA) of the retina in a group of PACD.

## 2. Materials and Methods

This study was a cross-sectional study enrolling patients with a diagnosis of PACD who visited the Zhongshan Ophthalmic Center. The study was approved by the institutional review board of the Zhongshan Ophthalmic Center and was performed in accordance with the tenets of the Declaration of Helsinki.

### 2.1. Healthy Control Subjects

Healthy control subjects were defined as follows: (1) aged between 40 and 80 years without obvious refractive media opacity that might affect imaging quality; (2) IOP less than 21 mmHg with no history of increased IOP; (3) normal anterior and posterior segment on clinical examination; and (4) an absence of a glaucomatous disc appearance and a retinal nerve fiber layer (RNFL) thickness within the normal range and normal visual field (VF);(5) without a history of trauma or other retinal or serve eye diseases.

### 2.2. PACD Patients

The inclusion criteria for patients with PACD were based on the diagnosis of PACD from Foster [5].

Patients with an invisible trabecular meshwork of more than 180° in circumference on static gonioscopy but without elevated IOP (>21 mmHg), peripheral anterior synechiae (PAS) or GON (defined as vertical cup to disc ratio > 0.7 or asymmetry > 0.2) were assigned to the PACS group.

PAC was defined as an eye with normal optic nerves and VF meeting gonioscopic criteria for occludable drainage angles (defined as an invisible pigmented trabecular meshwork of at least 180° on indentation gonioscopy in the primary position) with features indicating peripheral anterior synechiae (PAS) and no elevated IOP history.

APAC was defined as eyes with at least two of the following three symptoms: (1) ocular or periocular pain; (2) nausea, vomiting, or both; (3) an antecedent history of intermittent blurring of vision with haloes. The eyes should also have a history of elevated IOP and at least 3 of the following 4 signs: conjunctival injection, corneal epithelial edema, mid-dilated unreactive pupil, and shallow anterior chamber [17].

PACG was defined as PAC with glaucomatous optic neuropathy and compatible VF loss on static automated perimetry. The IOP of all of the PACD patients was controlled normally by glaucoma eye drops or anti-glaucoma surgery such as phacoemulsification surgery, trabeculectomy, or peripheral iridotomy.

All enrolled patients underwent thorough ophthalmic examinations, including best-corrected visual acuity (BCVA, measured by Snellen chart), IOP measurement, slit-lamp examination, fundus examination, gonioscopy, VF examinations, retinal nerve fiber layer (RNFL), ganglion cell complex (GCC) thickness, vessel density measurement, and blood FA measurement. The demographical and clinical characteristics of the participants were recorded. The study required patients have a BCVA of more than 0.1 (20/200) by the standard logarithmic visual acuity charts.

IOP was measured by Goldmann applanation tonometry. Gonioscopy was performed with a Goldmann 2-mirror lens by glaucoma specialists. The Swedish interactive threshold algorithm standard 30-2 strategy was used to examine VF (Carl Zeiss Meditec, Dublin, CA, USA). Only reliable tests (fixation losses < 20%, false positives < 33%, and/or false negatives < 33%) were included. The RNFL, GCC thickness, VD, and blood FA were assessed with spectral-domain optical coherence tomography (SD-OCT, RTVue-XR Avanti; Optovue, Inc., Fremont, CA, USA).

### 2.3. OCTA Examination

OCTA of all study eyes was imaged by a single experienced technician. The scans were obtained using the AngioVue software SD-OCT system Avanti RTVue XR (software version 2.0.5.39). Each raster scan consisted of 304 B-scans. Each B-scan consisted of 304 A-scans with an A-scan rate of 70,000 A-scans per second, and each raster scan took less than 3 s. Two volumetric raster scans, including 1 horizontal priority (x-fast) and 1 vertical priority (y-fast), were obtained consecutively within 15 s of each other. The volumetric scans were processed by the SSAD algorithm, and 3D orthogonal registration and the merging of 2 scans removed most motion artifacts [18].

### 2.4. Vessel Densities

The OCT-A scan within the peripapillary (4.5 × 4.5 mm^2^) and macular (6 × 6 mm^2^) areas. VD is defined as the percentage area occupied by the large vessels and microvasculature in a specific area.

In the scan, whole image vessel density (wiVD) was measured in the entire 4.5 ×4.5 mm^2^ image. The inside disc vessel density (idVD) was referred to as the average VD within the ONH, and peripapillary VD (pVD) was calculated in the region defined as a 750-μm-wide elliptical annulus extending from the optic disc boundary. The peripapillary area was subdivided into 6 sectors, including nasal, temporal, superonasal, superotemporal, inferonasal, and inferotemporal [19,20]. The ONH vessel densities were calculated from the “nerve head segment (extends from 150 μm below inner limiting membrane) to the “Radial Peripapillary Capillary (RPC) segment” (extends from the inner limiting membrane to RNFL).

Macular wiVD was calculated over the entire 6 × 6 mm^2^ scan field. Parafovea vessel densities (pfVD) were defined as an annulus with an outer diameter of 3 mm. Fovea vessel densities (fVD) were defined as the inner diameter of 1 mm centered at the fovea. The parafovea area was subdivided into 2 hemispheres (superior hemi and inferior hemi areas) and 4 sectors (superior, nasal, temporal, and inferior). The vessels were examined with a 4 plex analysis as superficial (from inner limiting membrane to 15 μm below inner plexiform layer), deep (15–70 μm below inner plexiform layer), outer retina (from 70 μm below inner plexiform layer to 60 μm below retinal pigment epithelium), and choroid cap (≥75 μm).

### 2.5. Blood Flow Area

Blood FA was defined as the area (mm^2^) occupied by the vessels in a specific area and reported in mm^2^. The blood FA of the ONH was measured in a circle with a 1 mm radius that was centered on the optic nerve head. The blood FA of macular was measured in selected areas (all 3.144 mm^2^) of 4 layers as superficial (from inner limiting membrane to 15 μm below IPL), deep (15–70 μm below IPL), outer retina (from 70 μm below IPL to 60 μm below RPE), and choroid cap (≥75 μm).

The exclusion criteria for the OCTA scans were signal strength index (SSI) < 50 or a scan containing severe artifacts, low quality of images with severe artifacts because of poor fixation, and layer segmentation errors (provided automatically by the software).

### 2.6. Statistical Analysis

All analyses were performed using statistical software (SPSS for Windows, version 22.0; SPSS Inc., Chicago, IL, USA). The Shapiro–Wilk test was used to evaluate the normal distribution of continuous variables. Descriptive statistics were calculated as the mean standard deviation for normally distributed variables, and medians (interquartile range) for nonnormally distributed variables.

Comparisons between all groups were performed using X^2^ test for categorical variables and by a Kruskal–Wallis test with post hoc Bonferroni correction for numerical data. The correlation between VD and FA in ONH or retinal and individual glaucomatous damage measurements was assessed using Spearman correlation analysis. *p*-values < 0.05 were considered statistically significant.

## 3. Results

Overall, 147 eyes from 121 subjects (31 eyes with APAC, 33 eyes with PACG, 25 eyes with PAC, 23 eyes with PACS, and 35 normal eyes) were included in the study. The demographical and clinical characteristics of the participants are listed in Table 1. There were no between-group differences in age, sex distribution, or IOP.

Among APAC eyes, 20 had undergone phacoemulsification surgery, and 11 had undergone trabeculectomy. All PACG eyes had undergone trabeculectomy. All PAC and PACS eyes had previously undergone peripheral iridotomy. The time interval between the surgery time and enrollment in this research ranged from 3 months to 1 year.

There was a significant difference among the five groups in terms of glaucoma parameters (*p* < 0.01). The VF, MD, RNFL, and GCC thicknesses were significantly lower in PACG and APAC than those in the other three groups (all *p* < 0.05). 

The amounts of peripapillary VD in different parts of the disc in the nerve head layer and RPC layer are summarized in Table 2. WiVD, idVD, pVD, and all sectors (nasal, temporal, superonasal, superotemporal, inferonasal, and inferotemporal) of pVD were significantly lower in both layers in eyes with PACG and APAC when compared to the those in the normal group, PAC group or PACS group (all *p* < 0.005) (Figure 1).

The blood FA was measured in a circle with a 1 mm radius centered on the optic nerve head with the same selected area (3.143 mm^2^) and reported in mm^2^ (Figure 2). The blood FA of the nerve head, vitreous, RPC and choroid are shown in Table 3. PACG and APAC eyes showed significantly decreased FA in the nerve head, vitreous and RPC layers (*p* < 0.001). There was no significant difference in the FA in all layers among PACS, PAC, and normal control groups or in the choroid layer among all groups (*p* > 0.05).

The macular wiVDs of 4 slabs were imaged (Figure 3) and the results are summarized in Table 4. The VDs of the superficial layers in different sectors are summarized in Table 4. The wiVD of the superficial layer in PACG or APAC eyes was significantly lower than that in other groups (*p* < 0.05). APAC and PACG eyes show a statistically decline in the superficial VD of the inferior-hemi, temporal and inferior areas (*p* < 0.01). There was also no significant difference among all groups in the wiVDs of the deep, outer retina and choroid cap layers nor in other sectors.

The macular blood FA of the superficial, deep, outer retina and choroid cap were imaged (Figure 4) and summarized in Table 5. Compared to control eyes, PACG and APAC eyes significantly decreased in the superficial layers but did not change in the deeper layers. No statistically significant difference in FA was found in the deep plex, outer retina, and choroid cap among all the groups.

Spearman correlation analysis is demonstrated in Figure 5 and Figure 6. MD, RNFL and GCC thicknesses were positively correlated with the VD and FA of the optic nerve.

## 4. Discussion

OCTA is a newly developed and noninvasive ocular imaging technique for evaluating many retinal and optic nerve diseases [7].

Previous studies of OCTA have demonstrated the decreased blood supply in both the optic nerve and peripapillary retina of POAG eyes, and the correlation between decreased vessel density and disease severity has been confirmed [7,8,9,11,12,21,22,23].

However, to our knowledge, there is no systematic research using OCTA to evaluate both vessel densities and blood FA in different types of PACD. Our study assessed superficial to deep capillary beds and calculated the vascular parameters with four slabs from the inner limiting membrane to the RPE.

Our results found a statistically significant lower VD of the ONH, peripapillary, and inside the disc of different layers of eyes with PACG and APAC than that of normal control eyes. Although the mean VD of each layer of PACG eyes was significantly lower than that of APAC eyes, there was no significant difference between the two groups. These results are consistent with previous reports that showed differences in OCT-A microvasculature between angle-closure glaucoma and healthy groups in the optic disc and peripapillary region. To date, only a few studies have observed the VD of PACD, but these studies did not include all types of PACD [24,25,26,27]. A few investigations have identified the distinct degrees and patterns of VD change in PACG or APAC eyes versus POAG eyes [15,28,29]. The diagnostic ability of peripapillary VD in evaluating glaucoma has been confirmed by several previous studies [13,30]. Vascular impairment was found in acute PACG eyes and exactly coincided with RNFL and GCC thicknesses [14,25,30]. L. RAO evaluated the VD of the optic nerve head (ONH) and peripapillary in APAC and PACG eyes and found that the VD was significantly lower in PACG and APAC eyes than in control eyes [15]. We also observed, for the first time, normal VD in patients with PACS and PAC, who demonstrated no optic nerve damage. Studies on POAG have shown that changes in VD on optic discs occur earlier than VF damage, and their test power (AUC) (0.94) is even slightly greater than the thickness of the nerve fiber layer (0.92) [10]. Taken together, our findings demonstrated that the decreased VD of the ONH, peripapillary, and inside discs was an important morphological parameter for the evaluation of damage in patients with PACD.

The study of FA showed a similar result that the FA of ONH in APAC and PACG eyes were statistically diminished in the ONH, RPC and vitreous layers compared with normal control eyes. However, no significant difference was found in the choroid layer among the five groups of subjects. To our knowledge, no research has measured the blood FA of the ONH in all kinds of PACD [26]. The analysis suggests that the blood FA can provide a reference for the evaluation of blood perfusion of the ONH in PACD patients.

Histological studies have shown that glaucoma leads to a reduction in retinal ganglion cells (RGCs), which is one of the pathophysiological bases of glaucoma. Nearly 50% of RGCs are distributed in the macula area [31,32]. Therefore, the thickness measurement of the macula is of a certain value for the evaluation of glaucomatous optic nerve injury [33]. Relevant studies of OCT show that local lesions of GCC may be a good indicator of glaucoma progression. Therefore, it is necessary to pay attention to studies of the macular area in the diagnosis and monitoring of glaucoma [34]. However, the glaucoma-induced damage to macular circulation has not yet been confirmed.

In the macular scan, we found for the first time, evidence of a significant decline in the wiVD and FA of the superficial layer, but not in the deep, outer retina and choroid cap layers. In addition, statistically significant declines were found in the superficial VD of the inferior-hemi, temporal and inferior areas of APAC and PACG eyes but not in the fovea, parafovea, superior-hemi, superior, nasal sectors. Our result was consistent with the studies concerning the macular area in PACG and PAC eyes [15,29]. Rao et al. [15] found lower retinal vessel densities in the macular scan of PACG eyes but not PAC eyes. Our result is also similar to the studies of POAG, which documented vascular damage in the temporal regions of the superficial macula in POAG eyes [15,35,36,37]. Since PACS and PAC might partly develop into the APAC or PACG, it was quite necessary to monitor the VD change especially macular VD to distinguish the VD change in the PACS or PAC stage. We also found both VD and flow area did not statistically differ between PAC or PACS and the normal control group. This result is consistent with the findings of Zha’s [26] study. They found no statistically significant difference in macular VD between PACS and the normal control. The reason for our result may be that PAC and PACS present only preclinical anatomical abnormalities, such as an anterior chamber with a normal optic nerve. Therefore, no detectable change in VD or FA was found.

Moreover, we found that the decline in the superficial macular VD in APAC and PACG eyes is located in the temporal and inferior areas. These results were consistent with the previous understanding of glaucomatous optic nerve damage, suggesting that glaucoma mainly damages the temporal and inferior areas. This finding suggested that acute or chronic elevated IOP could lead to vascular changes in the temporal and inferior areas of the superficial macula.

Furthermore, we simultaneously observed the relationship between traditional glaucoma parameters and VD and FA in OCTA. Three indicators, including VF MD value, RNFL, and GCC thickness, which can accurately represent the severity of glaucoma damage, were selected for correlation analysis with the parameters obtained from OCTA. This analysis showed that the pVD (nerve head layer and RPC layer) and blood FA of the nerve head layer were the most highly correlated with MD, RNFL, and GCC thickness, while the wViD and blood FA in the vitreous also showed a positive correlation with these parameters. However, no correlation was found between the iVD or blood FA of the choroid layer and the three glaucoma parameters. These findings are similar to previous studies on POAG and PACG eyes, where the peripapillary microcirculation was highly correlated with VF, MD, RNFL, and GCC thickness [14,38].

VF, MD, RNFL, and GCC have been clinically recognized as the most important parameters for the measurement of glaucomatous impairment and the severity of the disease. The findings in the present study show that the altered wiVD and pVD of the nerve head layer and RPC, blood FA of the nerve head layer, RPC layer, and vitreous are associated with the severity of glaucomatous damage in both PACG and APAC eyes. These data suggest that angio-OCT disc blood flow measurements are good indicators of eyes with PACG and APAC. Although all indicators of angio-OCT disc blood flow measurements are consistent with the changes in other classical glaucomatous parameters, their diagnostic value was varied. Among the measurements, no correlation was found between either WiVD or FA of the choroidal layer and the other glaucomatous parameters, showing that the diagnostic value in PACD of these two indicators may be small. In contrast, as it is highly correlated with other glaucomatous parameters, the pVD of the nerve head layer can be an excellent indicator for examining the severity of APAC and PACG eyes. Therefore, much attention should be paid to the pVD in angio-OCT disc blood flow measurements over PACD.

There are a few limitations of the study. The main limitations of our study include a selection bias at baseline. The increase in intraocular pressure in APAC patients will lead to corneal edema and affect the quality of the image. Therefore, we do not include untreated eyes with PACD in our study. Additionally, to ensure good cooperation during VF recording and safety during pupil dilation, this study excludes patients with visual acuity lower than 0.1 and patients without surgical or laser treatment. This cross-sectional study limited the power of our findings. Longitudinal studies are needed to confirm our conclusion and to determine the influence of different surgical modalities and postoperative treatment on VD.

In summary, this study quantitatively evaluated the VD and FA of the optic nerve and macular using OCTA in PACD. There were decreased VD and FA in eyes with PACG and APAC compared with healthy eyes, and this change is consistent with the changes in other classical glaucomatous parameters, which can be used as a morphological indicator of PACD optic nerve damage observation.

## Figures and Tables

**Figure 1 jcm-11-04040-f001:**
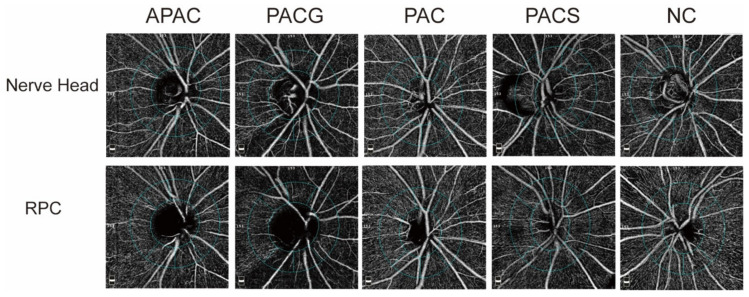
Image of vessel density (VD) in the nerve head layer and radial peripapillary capillary (RPC) layer. The vessel densities of peripapillary sectors are calculated over a 0.75 mm-wide elliptical annulus extending from the optic disc boundary. APAC = acute primary angle-closure; PACG = primary angle-closure glaucoma; PAC = primary angle-closure; PACS = primary angle-closure suspect; NC = normal control.

**Figure 2 jcm-11-04040-f002:**
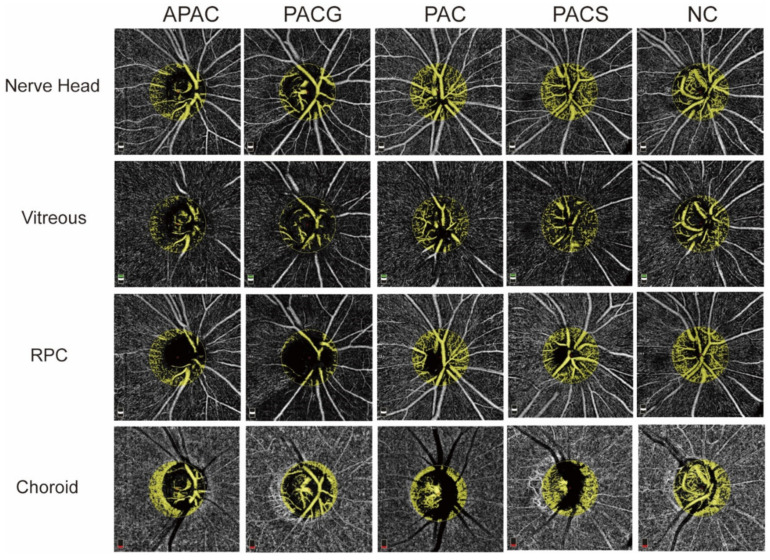
Image of the flow area (FA) of the optic disc. The FA was measured in the central 1 mm^2^ area of the disc. PACG and APAC eyes show significantly a decreased FA in the nerve head, vitreous, and RPC layers.

**Figure 3 jcm-11-04040-f003:**
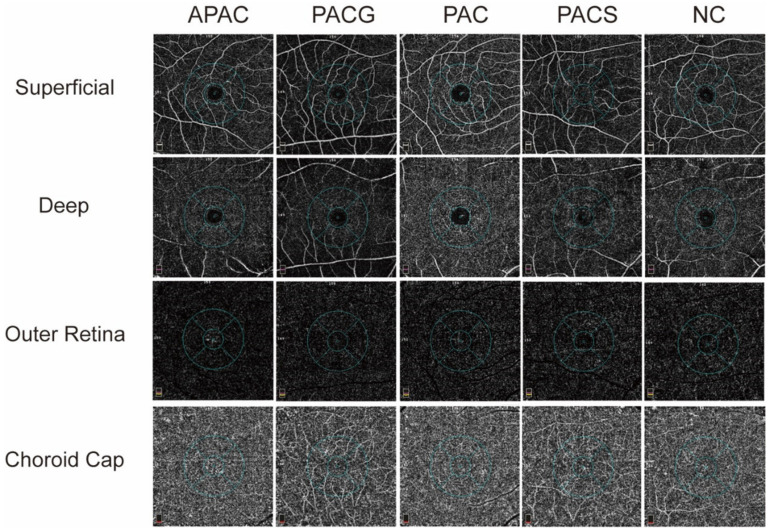
Image of the ocular vascular network in different retinal layers in primary angle-closure diseases (PACD) and healthy eyes. A comparison of the vascular density in the superficial, deep, outer retinal, and choroid cap layers in foveal and parafoveal areas are presented. APAC = acute primary angle-closure; PACG = primary angle-closure glaucoma; PAC = primary angle-closure; PACS = primary angle-closure suspect; NC = normal control.

**Figure 4 jcm-11-04040-f004:**
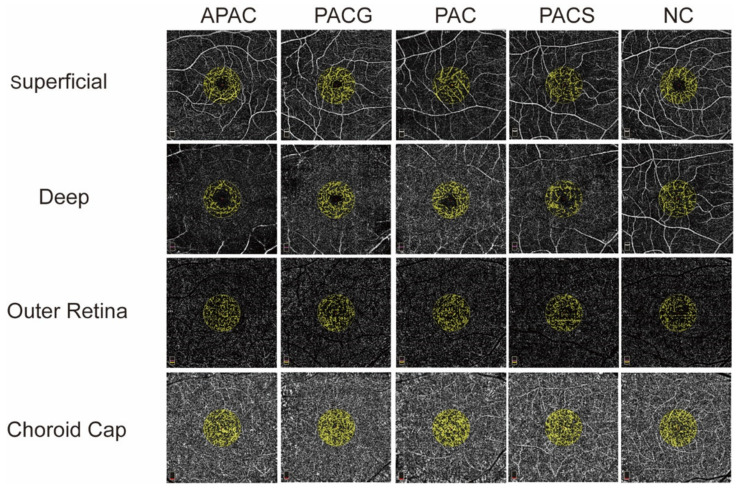
Image of macular flow area (FA). A comparison of the superficial, deep, outer retina, and choroid cap layers is presented. The FA was measured in a 1-mm radius of the foveal area (6 mm^2^ × 6 mm^2^) with a preset setting. APAC = acute primary angle-closure; PACG = primary angle-closure glaucoma; PAC = primary angle-closure; PACS = primary angle-closure suspect; NC = normal control.

**Figure 5 jcm-11-04040-f005:**
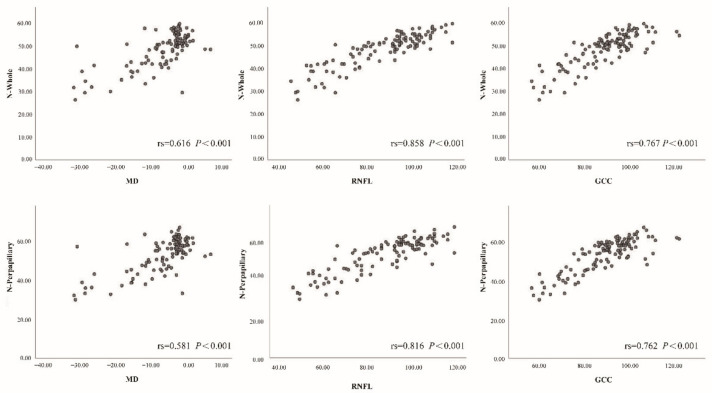
Relationships of whole image vessel density (N-Whole) and peripapillary vessel density (N-Perpapillary) with VF mean deviation (MD), mean retinal nerve fiber layer (RNLF), and ganglion cell complex thickness (GCC) in all objects.

**Figure 6 jcm-11-04040-f006:**
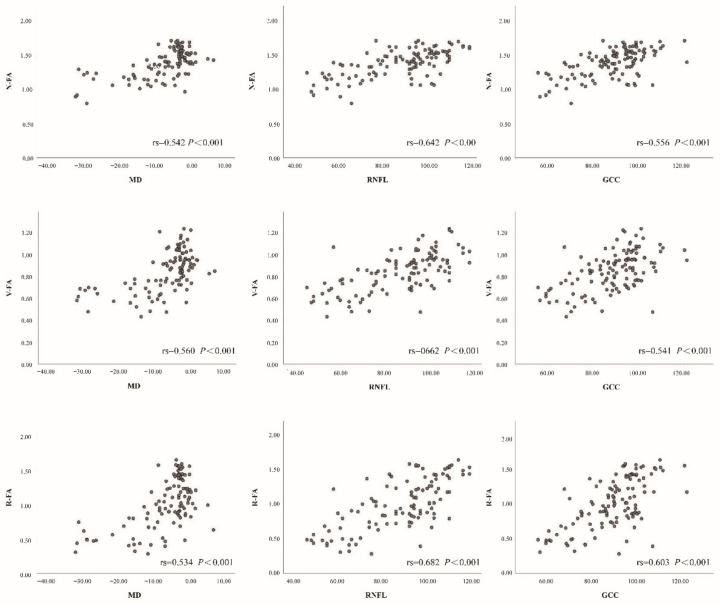
Relationships of flow area of nerve head (NFA), flow area of vitreous (VFA), and flow area of radial peripapillary capillary (VFA) with VF mean deviation (MD), mean retinal nerve fiber layer (RNLF) and ganglion cell complex thickness (GCC) in all objects.

**Table 1 jcm-11-04040-t001:** Demographical and clinical characteristics of the participants.

	APAC (*n* = 31)	PACG (*n* = 33)	PAC (*n* = 25)	PACS (*n* = 23)	NC (*n* = 35)	P^1^	P^2^	P^3^	P^4^
Age	59.67 ± 8.38	60.86 ± 7.90	58.87 ± 6.25	59.81 ± 10.65	58.66 ± 10.57	*p* = 0.108			
Sex (M:F)	6:29	5:28	3:22	4:19	7:28				
IOP (mmHg)	12 (10.9, 14.2)	13.0 (11.0, 16.2)	13.3 (12.1, 14.5)	13.5 (12.1, 17.0)	13.8 (11.8, 15.1)	*p* = 0.456			
MD (dB)	−11.48 (−7.72, 2.22)	−11.69 (−28.3, −6.76)	−2.05 (−3.07, 1.07)	−2.42 (−3.37, 1.52)	−2.62 (−3.53, −0.06)	<0.001	0.054	0.737	0.754
PSD (dB)	3.32 (2.04, 8.32)	6.14 (2.04, 8.04)	1.69 (1.46, 2.07)	1.79 (1.38, 2.69)	1.79 (1.35, 2.40)	0.009	<0.001	0.937	0.872
RNFL (μm)	76.50 (64.25, 93.00)	69.50 (56.00, 95.50)	101.50 (95.75, 109.25)	102.00 (91.75, 108.25)	98.31 ± 10.19	<0.001	<0.001	0.492	0.793
GCCth (μm)	81.18 (72.00, 92.86)	75.43 (63.93, 89.15)	96.18 (91.81, 101.35)	97.94 (92.76, 104.35)	95.15 ± 8.58	0.009	<0.001	0.731	0.262
Trabeculectomy (eyes)	23	33	0	0	0				
Phacoemulsification (eyes)	12	0	0	0	0				
Peripheral iridectomy (eyes)	6	0	19	23	0				

APAC = acute primary angle-closure; PACG = primary angle-closure glaucoma; PAC = primary angle-closure; PACS = primary angle-closure suspect; NC = normal control; IOP = intraocular pressure. *p* represents the *p*-value associated with the comparisons among all groups. P^1^ represents the *p*-value associated with the comparisons between control and APAC groups. P^2^ represents the *p*-value associated with the comparisons between control and PACG groups. P^3^ represents the *p*-value associated with the comparisons between control and PAC groups. P^4^ represents the *p*-value associated with the comparisons between control and PACS groups.

**Table 2 jcm-11-04040-t002:** Vessel density in peripapillary sectors of the nerve head layer and radial peripapillary capillary layer.

	APAC (*n* = 35)	PACG (*n* = 33)	PAC (*n* = 25)	PACS (*n* = 23)	NC (*n* = 35)	P^1^	P^2^	P^3^	P^4^
Nerve head-whole image (%)	48.18 (40.91, 50.35)	42.21 (35.90, 48.75)	53.49 (51.47, 56.94)	52.06 (49.76, 56.06)	51.85 (50.34, 54.99)	<0.001	<0.001	0.54	0.86
Perpapillary (%)	52.61 ± 3.18	45.58 ± 9.23	58.13 ± 4.89	56.60 ± 5.02	57.35 ± 3.43	<0.001	0.002	0.70	0.59
inside disc (%)	43.19 (36.64, 46.70)	40.12 (34.03, 45.04)	49.27 (45.04, 52.32)	47.98 (41.68, 51.53)	46.72 (43.76, 51.14)	0.003	<0.001	0.40	0.96
Nasal (%)	48.05 (38.50, 53.05)	43.40 (36.39, 51.03)	55.08 (51.83, 60.60	53.96 (50.38, 57.98)	55.87 (52.07, 57.94)	0.001	<0.001	0.79	0.85
Inferonasal (%)	49.66 (45.49, 56.46)	46.76 (42.13, 52.66)	60.05 (54.49, 62.37)	59.03 (54.22, 63.66)	59.81 (55.72, 62.04)	<0.001	<0.001	0.95	0.80
Inferotemporal (%)	56.73 (42.92, 61.21)	48.94 (38.04, 60.99)	61.60 (58.64, 66.48)	62.39 (59.73, 64.69)	63.62 (59.97, 66.28)	<0.001	<0.001	0.69	0.74
Superotemporal (%)	54.03 (42.70, 58.76)	47.33 (34.67, 55.57)	61.44 (57.81, 66.03)	61.44 (57.81, 66.03)	61.53 (57.91, 64.42)	<0.001	<0.001	0.99	0.46
Superonasal (%)	49.06 (40.95, 54.83)	43.99 (38.26, 50.48)	56.30 (54.32, 61.42)	56.29 (51.56, 61.64)	57.03 (54.13, 60.08)	0.004	<0.001	0.97	0.97
Temporal (%)	52.51 (43.84, 56.21)	49.09 (38.26, 56.54)	59.42 (56.63, 63.79)	53.44 (49.39, 60.03)	58.61 (54.22, 60.31)	0.004	0.003	0.31	0.27
RPC-whole image (%)	44.11 (37.62, 47.76)	39.17 (30.70, 48.37)	51.17 (49.46, 56.68)	52.85 (48.48, 52.02)	51.50 (50.48, 53.24)	<0.001	<0.001	0.96	0.72
Perpapillary (%)	52.39 (44.96, 56.23)	48.39 (40.85, 54.30)	59.66 (56.70, 61.90)	58.44 (54.83, 62.46)	60.87 (58.07, 62.67)	<0.001	<0.001	0.58	0.28
inside disc (%)	23.95 (16.85, 35.21)	19.52 (11.04, 27.62)	37.90 (24.55, 46.85)	40.20 (32.19, 44.96)	38.48 (34.42, 43.14)	<0.001	<0.001	0.62	0.92
Nasal (%)	47.33 (41.79, 52.82)	43.87 (34.28, 50.86)	57.66 (50.59, 60.03)	55.91 (51.70, 59.94)	57.2 (55.10, 60.82)	<0.001	<0.001	0.75	0.54
Inferonasal (%)	50.16 (43.56, 56.08)	47.00 (39.66, 54.48)	58.98 (52.67, 61.70)	60.98 (58.93, 63.00)	60.60 (58.03, 63.56)	<0.001	<0.001	0.34	0.80
Inferotemporal (%)	57.18 (45.74, 61.73)	51.44 (35.8, 61.01)	63.74 (60.30, 67.57)	64.04 (60.70, 62.29)	66.55 (62.37, 69.37)	<0.001	<0.001	0.37	0.57
Superotemporal (%)	54.78 (45.89, 60.58)	44.34 (34.38, 57.49)	65.30 (57.68, 67.26)	63.13 (57.92, 65.90)	64.30 (61.64, 66.14)	<0.001	<0.001	0.82	0.38
Superonasal (%)	48.42 (41.50, 52.72)	41.92 (37.07, 53.36)	57.18 (53.46, 59.45)	58.69 (52.75, 60.20)	58.81 (55.67, 62.76)	<0.001	<0.001	0.32	0.41
temporal (%)	54.71 (50.79, 50.08)	52.38 (39.36, 59.39)	62.87 (60.05, 65.36)	57.10 (53.37, 61.26)	62.14 (58.00, 64.38)	0.002	0.001	0.54	0.71

APAC = acute primary angle-closure; PACG = primary angle-closure glaucoma; PAC = primary angle-closure; PACS = primary angle-closure suspect; IO All values are shown as the means ± standard deviations. P^1^ represents the *p*-value associated with the comparisons between control and APAC groups. P^2^ represents the *p*-value associated with the comparisons between control and PACG groups. P^3^ represents the *p*-value associated with the comparisons between control and PAC groups. P^4^ represents the *p*-value associated with the comparisons between control and PACS groups.

**Table 3 jcm-11-04040-t003:** The flow area (FA) of retina.

	APAC (*n* = 31)	PACG (*n* = 33)	PAC (*n* = 25)	PACS (*n* = 23)	NC (*n* = 35)	P^1^	P^2^	P^3^	P^4^
Optic Disc									
Nerve head-FA (mm^2^)	1.27 (1.16, 1.42)	1.26 (1.14, 1.40)	1.57 (1.47, 1.67)	1.49 (1.39, 1.56)	1.49 (1.37, 1.57)	0.008	0.004	0.44	0.95
Vitreous-FA (mm^2^)	0.74 (0.62, 0.88)	0.69 (0.62, 0.85)	0.96 (0.93, 1.03)	0.89 (0.38, 1.04)	0.93 (0.84, 1.04)	0.002	<0.001	0.39	0.71
RPC-FA (mm^2^)	0.89 (0.70, 1.150)	0.72 (0.48, 0.94)	1.35 (0.96, 1.52)	1.34 (1.07, 1.43)	1.03 (0.87, 1.38)	0.05	0.01	0.11	0.10
Choroid-FA (mm^2^)	1.49 (1.34, 1.59)	1.53 (1.42, 1.59)	1.44 (1.24, 1.63)	1.37 (1.26, 1.49)	1.56 (1.39, 1.61)	*p* = 0.14			
Macular									
Superficial	1.25 (1.15, 1.40)	1.33 (1.15, 1.46)	1.60 (1.43, 1.69)	1.51 (1.35, 1.56)	1.51 (1.35, 1.58)	*p* = 0.53			
Deep	0.72 (0.62.0.93)	0.77 (0.64, 0.91)	0.95 (0.77, 1.04)	0.88 (0.83, 1.04)	0.93 (0.83, 1.04)	*p* = 0.27			
Outer Retina	0.86 (0.62, 1.18)	0.82 (0.49, 1.10)	1.35 (0.86, 1.55)	1.21 (1.05, 1.42)	1.10 (0.88, 1.41)	*p* = 0.22			
Choroid Cap	1.49 (1.33, 1.59)	1.54 (1.46, 1.59)	1.43 (1.21, 1.55)	1.39 (1.33, 1.52)	1.51 (1.30, 1.59)	*p* = 0.33			

APAC = acute primary angle-closure; PACG = primary angle-closure glaucoma; PAC = primary angle-closure; PACS = primary angle-closure suspect; NC = normal control; IOP = intraocular pressure. All values are shown as the means ± standard deviations. P^1^ represents the *p*-value associated with the comparisons between control and APAC groups. P^2^ represents the *p*-value associated with the comparisons between control and PACG groups. P^3^ represents the *p*-value associated with the comparisons between control and PAC groups. P^4^ represents the *p*-value associated with the comparisons between control and PACS groups.

**Table 4 jcm-11-04040-t004:** Macular whole image vessel density (wiVD).

	APAC (*n* = 31)	PACG (*n* = 33)	PAC (*n* = 25)	PACS (*n* = 23)	NC (*n* = 35)	P^1^	P^2^	P^3^	P^4^
Superficial (%)	40.00 (37.72, 44.40)	39.99 (37.72, 44.40)	43.40 (41.25, 46.63)	45.02 (42.64, 47.08)	43.19 (40.92, 49.79)	0.59	0.51	0.82	0.77
Deep (%)	49.65 (45.62, 51.71)	49.65 (45.62, 51.71)	51.00 (48.76, 57.35)	51.61 (47.46, 53.34)	52.58 (48.88, 56.78)	0.87	0.12	0.68	0.43
Outer Retina (%)	45.15 (43.25, 46.64)	45.15 (43.43, 46.64)	44.80 (42.39, 45.72)	43.81 (41.82, 44.94)	43.73 (42.79, 44.32)	*p* = 0.29			
Choroid Cap (%)	63.43 (62.16, 63.67)	62.43 (62.16, 63.67)	63.43 (61.67, 64.78)	64.09 (62.75, 65.08)	62.77 (62.42, 64.34)	*p* = 0.30			

APAC = acute primary angle-closure; PACG = primary angle-closure glaucoma; PAC = primary angle-closure; PACS = primary angle-closure suspect; NC = normal control; IOP = intraocular pressure. *p* represents the *p*-value associated with the comparisons among all groups. P^1^ represents the *p*-value associated with the comparisons between control and APAC groups. P^2^ represents the *p*-value associated with the comparisons between control and PACG groups. P^3^ represents the *p*-value associated with the comparisons between control and PAC groups. P^4^ represents the *p*-value associated with the comparisons between control and PACS groups.

**Table 5 jcm-11-04040-t005:** Whole image vessel density (wiVD) of the superficial layer.

	APAC (*n* = 31)	PACG (*n* = 33)	PAC (*n* = 25)	PACS (*n* = 23)	NC (*n* = 35)	P^1^	P^2^	P^3^	P^4^
Fovea (%)	45.06 (41.91, 47.45)	45.06 (41.91, 47.45)	47.34 (41.31, 49.97)	47.67 (44.79, 50.61)	50.50 (44.06, 51.25)	*p* = 0.79			
ParaFovea (%)	25.73 (22.88, 30.89)	25.73 (22.88, 30.89)	24.18 (19.85, 28.69)	26.27 (23.20, 35.15)	31.19 (25.24, 36.97)	*p* = 0.15			
Superior-Hemi (%)	44.28 (42.66, 46.64)	45.02 (41.48, 47.86)	47.66 (41.91, 51.12)	46.56 (43.42, 48.71)	47.44 (44.08, 50.05)	0.14	0.07	0.64	0.69
Inferior-Hemi (%)	44.22 (41.22, 46.47)	43.22 (41.86, 45.84)	47.35 (44.99, 49.95)	47.94 (43.57, 49.47)	50.09 (43.47, 51, 16)	0.09	0.05	0.73	0.49
Tempo (%)	45.19 (42.93, 48.17)	45.61 (43.17, 47.62)	47.76 (45.73, 50.71)	48.77 (45.81, 49.72)	49.98 (47.08, 53.10)	0.002	0.001	0.32	0.28
Superior (%)	44.07 (40.97, 45.69)	45.05 (43.09, 46.51)	45.83 (40.56, 51.49)	46.06 (43.75, 49.29)	44.45 (43.62, 49.41)	0.23	0.15	0.76	0.97
Nasal (%)	45.17 (42.04, 48.15)	45.05 (43.09, 46.51)	46.95 (42.26, 49.74)	47.14 (44.38, 48.90)	45.72 (43.22, 51.52)	0.40	0.38	0.84	0.94
Inferior (%)	43.13 (39.95, 44.92)	43.33 (39.32, 46.21)	46.45 (42.63, 50.53)	47.40 (41.47, 50.66)	49.52 (47.79, 50.12)	0.16	0.13	0.42	0.69

APAC = acute primary angle-closure; PACG = primary angle-closure glaucoma; PAC = primary angle-closure; PACS = primary angle-closure suspect; IOP = intraocular pressure. *p* represents the *p*-value associated with the comparisons among all groups. P^1^ represents the *p*-value associated with the comparisons between control and APAC groups. P^2^ represents the *p*-value associated with the comparisons between control and PACG groups. P^3^ represents the *p*-value associated with the comparisons between control and PAC groups. P^4^ represents the *p*-value associated with the comparisons between control and PACS groups.

## Data Availability

The datasets generated and analyzed during the current study are available from the corresponding author on reasonable request.

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
