# Peer review of "Quantitative Measurements of Vessel Density and Blood Flow Areas Primary Angle Closure Diseases: A Study of Optical Coherence Tomography Angiography"

_jcm, 2022, doi:10.3390/jcm11144040_

Round 1

Reviewer 1 Report

I read the paper entitled »Quantitative measurements of vessel density and blood flow areas in primary angle closure disease: a study of optical coherence tomography angiography « very carefully and concluded that the paper is not acceptable for publication in your journal.

Title must be changed. “appears to have....what that meen? The authors concluded that it had.

Descriptions of statistical analysis of the data is not clear.. only paired t-test was performed?

In results, the number of eyes are wrong, the results must be the same regarding the text and  presented results in Table 1. The number of eyes must be present as a text in the results.

The differencies in the results must be changed.

Results: The analysis of the data is not appropriate  regarding the vessel density and blood flow areas in primary angle closure disease in  optical coherence tomography angiography is vious results and must be changed.

Discussion: some new references must be added regarding the vessel density and blood flow areas in primary angle closure disease in  optical coherence tomography angiography.

Table and figures: see comments in results, the  number musct be the same as in text.

References: some new references must be added especially from the last year.

Author Response

Point 1. Title must be changed. “appears to have....what that mean? The authors concluded that it had.

Response 1: Thank you for your valuable comments. We checked the title and full text carefully, but we did not find “appears to have”.The title of our paper is “Quantitative Measurements of Vessel Density and Blood Flow Areas Primary Angle Closure Diseases: A study of Optical Coherence Tomography Angiography

Point 2. Descriptions of statistical analysis of the data is not clear.. only paired t-test was performed?

Response 2: Thank you for your question. As the reviewer suggested, we rechecked the statistical analysis and found paired t-test was not performed in this study. We employed the Shapiro-Wilk test to evaluate the normal distribution of continuous variables, X² test for categorical variables and by a Kruskal-Wallis test with post hoc Bonferroni correction for numeiracal data. The correlation was assessed using Spearman correlation analysis. Statistical analysis is detailed described in the Methods section.

“Statistical analysis

All analyses were performed using statistical software (SPSS for Windows, version 22.0; SPSS Inc. Chicago, IL, USA). The Shapiro-Wilk test was used to evaluate the normal distribution of continuous variables. Descriptive statistics were calculated as the mean standard deviation for normally distributed variables,and medians(25th–75th quantile)  for nonnormally distributed variables.

Compassion between all groups were performed using X² test for categorical variables and by a Kruskal-Wallis test with post hoc Bonferroni correction for numeiracal data. The correlation between VD, FA in ONH or retinal and individual glaucomatous damage measurements was assessed using Spearman correlation analysis. P-values < 0.05 were considered statistically significant.”

Point 3. In results, the number of eyes are wrong, the results must be the same regarding the text and  presented results in Table 1. The number of eyes must be present as a text in the results. The differencies in the results must be changed.

Response 3:  We appreciate the careful reading of the paper. As the reviewer suggested, the number of eyes have been checked and modified accordingly.

Point 4. Results: The analysis of the data is not appropriate  regarding the vessel density and blood flow areas in primary angle closure disease in  optical coherence tomography angiography is vious results and must be changed.                                                                                                                                            Response 4:According to the suggestion of the reviewer, we have checked the statistical section.  We have changed Pearson analysis to Spearman analysis in the correlation analysis part.

Point 5. Discussion: some new references must be added regarding the vessel density and blood flow areas in primary angle closure disease in optical coherence tomography angiography.

Response 5:Thank you for your valuable advice. According to the suggestion, we have added latest references.

Point 6. Table and figures: see comments in results, the  number musct be the same as in text.

Response 6:We appreciate the careful reading of the paper. As the reviewer suggested, the number of eyes have been checked and modified accordingly.

Point 7. References: some new references must be added especially from the last year.

Response 7: Thank you for your valuable advice. According to the suggestion, we have added several latest references(references 25-28).

25. Liu K, Xu H, Jiang H, et al. Macular vessel density and foveal avascular zone parameters in patients after acute primary angle closure determined by OCT angiography. Scientific reports. 2020;10(1):18717.

26.Zha Y, Chen J, Liu S, et al. Vessel Density and Structural Measurements in Primary Angle-Closure Suspect Glaucoma Using Optical Coherence Tomography Angiography. Biomed Res Int. 2020;2020:7526185.

27.Lin Y, Chen S, Zhang M. Peripapillary vessel density measurement of quadrant and clock-hour sectors in primary angle closure glaucoma using optical coherence tomography angiography. BMC Ophthalmol. 2021;21(1):328.

28. Suwan Y, Fard MA, Petpiroon P, et al. Peripapillary Perfused Capillary Density in Acute Angle-Closure Glaucoma: An Optical Coherence Tomography Angiography Study. Asia Pac J Ophthalmol (Phila). 2021;10(2):167-172.

Reviewer 2 Report

Dear Author(s),

Thanks for your submission on the JCM.

I have read with interest the microvascular abnormalities found in your cohort of patients affected by primary open angle suspect, primary angle closure, primary angle closure glaucoma. Indeed, in literature only small studies about the application of OCTA in a particular setting, as PACG, are found so your report resulted more complete. For the vessel density analysis performed, I would like to ask you why you did not binarise the images before calculating this parameter. This is the common way in which vessel density is obtained. Furthermore, the article needs a profound formal revision. In many parts, the speech results redundant or confusing, in others the language should be carefully revise. 

Author Response

Point 1. For the vessel density analysis performed, I would like to ask you why you did not binarise the images before calculating this parameter. This is the common way in which vessel density is obtained.

Response 1:Thank you for the question. As the reviewer noted, there were some researchers binarise the images to calculate vessel density some literatures1-2. We use Avanti RTVue XR (software version 2.0.5.39) to get vessel density in our research. The vessel density was automatically calculated by the software. Therefore, we didn’t binarise the images. The repeatability and reproducibility of the Optovue Avanti were previously published3-5.

1.Costanzo E, Parravano M, Giannini D, et al. Imaging Biomarkers of 1-Year Activity in Type 1 Macular Neovascularization. Translational vision science & technology. 2021;10(6):18.

2.Ashraf M, Nesper PL, Jampol LM, et al. Statistical Model of Optical Coherence Tomography Angiography Parameters That Correlate With Severity of Diabetic Retinopathy. Invest Ophthalmol Vis Sci. 2018;59(10):4292-4298.

3.Samara WA, Shahlaee A, Adam MK, et al. Quantification of Diabetic Macular Ischemia Using Optical Coherence Tomography Angiography and Its Relationship with Visual Acuity. Ophthalmology. 2017;124(2):235-244.

4.Martucci A, Giannini C, Di Marino M, et al. Evaluation of putative differences in vessel density and flow area in normal tension and high-pressure glaucoma using OCT-angiography. Progress in brain research. 2020;257:85-98.

5.Xu H, Zhai R, Zong Y, et al. Comparison of retinal microvascular changes in eyes with high-tension glaucoma or normal-tension glaucoma: a quantitative optic coherence tomography angiographic study. Graefes Arch Clin Exp Ophthalmol. 2018;256(6):1179-1186.

Point 2. Furthermore, the article needs a profound formal revision. In many parts, the speech results redundant or confusing, in others the language should be carefully revise.

Response 2:We appreciate the close review of the paper. We have rewritten and streamlined the result part of the manuscript.

Round 2

Reviewer 2 Report

Dear Author(s),

After the due revision made in accordance with the reviewer's suggestions, I would now recommend this paper for publication. Congratulations!

This manuscript is a resubmission of an earlier submission. The following is a list of the peer review reports and author responses from that submission.